# Consumers' Motives on Wine Tourism in Greece in the Post-COVID-19 Era

Athanasios Santorinaios, Ioanna S. Kosma 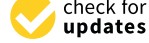 and Dimitris Skalkos *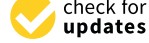

Laboratory of Food Chemistry, Department of Chemistry, University of Ioannina, 45110 Ioannina, Greece; ch06054@uoi.gr (A.S.); i.kosma@uoi.gr (I.S.K.)
* Correspondence: dskalkos@uoi.gr; Tel.: +30-2651008345

**Abstract:** Wine tourism is emerging as one of the most important forms of alternative, sustainable tourism in wine countries, such as Greece, in the post-COVID-19 era. In this paper, consumers' motives for wine tourism in Greece today are investigated regarding (i) their consumption habits related to wine, (ii) their experience with wine tourism, (iii) the parameters that would encourage their visit to a wine region, such as wine, the winery, and general regional characteristics, and (iv) the source of information consulted for a wine tourism experience. The questionnaire was conducted from April to May 2023, with 595 participants, via the Google Forms platform. The statistical analysis was performed with basic tools, as well as cross and chi-square tests, to analyze the data. The highlights of the results indicate that consumers (the participants of the survey) consume more wine today than before the pandemic (57%) and have previous experience in wine tourism (59.8%), with the majority of them having visited a winery more than once (67.4%). The most popular activity at the winery was found to be wine tasting (46.6%), followed by open discussion about wine (35.2%), and, at the regional level, visiting the sights (46%) and doing activities in nature (30.6%). Future participants are looking for innovation in wine tourism, with trained staff (77.5%) and organized tours (74.3%), the organization of wine festivals and other events (71.9%), opportunities to explore the local community, such as the outdoors (83.5%) and its culture and history (70.9%), during their visit, and available information on wine tourism opportunities online (73%). They also are encouraging the transition of the Greek wine tourism industry to the digital world. Based on the overall results, three types of support are proposed for the successful, sustainable development of wine tourism in wine-producing countries.

**Keywords:** questionnaire survey; wine tourism; post-COVID-19; Greece

## 1. Introduction

The COVID-19 pandemic brought significant changes to the tourism industry, with travelers reconsidering their goals and intentions for their trips [1]. Sustainability has emerged as the new trend in tourism in the post-pandemic era [2], with alternative tourism potentially replacing parts of mass tourism as part of its resurgence in the current times [3]. Wine tourism highlights its sustainability, as demonstrated by the economic and cultural considerations that have been perceived by the local communities [4]. For example, wine tourists' satisfaction is enhanced when their winery visit includes elements of local cuisine and wine culture [5]. With the discovery of grape wine being recorded from the beginning of the sixth millennium BC [6], wine production has a history of thousands of years, having survived detrimental climate changes of the past [7]. However, wine consumption has been always linked to a better quality of life, especially in old age, as well as low mortality [8]. Wine-producing countries are also now focusing more on aesthetic and experiential consumer satisfaction, rather than exclusively on knowledge-sharing about winemaking and the cellar's facilities [9]. This is a result of the similarities between wine consumption and art appreciation that have been observed from the consumer's

perspective [10]. Wine tourism can promote local wine more effectively, boosting its exports and sales [11]. The aim of this research is to identify the parameters that will boost wine tourism in a wine-producing country during the post-pandemic period and beyond, supporting regional sustainable development for the local communities and wineries benefiting from this innovative, modern, and future tourism industry. To achieve this aim the relevant literature regarding wine tourism was systematically reviewed and presented, followed by the materials and method of the study, the results, and the findings.

*Literature Review*

The COVID-19 pandemic greatly affected all tourism sectors in Europe and Greece [12], including wine tourism. Wine tourists in the near and far future will demand safer, healthier, and more sustainable experiences [13]. After the COVID-19 pandemic, it appears that visitors prefer less popular and crowded destinations [14]. Especially for domestic travelers, tranquility and rest, elements directly linked to wine tourism through the enjoyment of the natural landscape, have been shown to be key motivators for their visit to the country's wine regions [15]. In a psychophysiology study, higher levels of satisfaction were recorded when participants were able to freely explore different areas of the winery, compared to when they were given explanations [16]. Generation Y (born between 1981 and 1994), for example, the largest group of wine tourists, have a pleasant desire to consume wine if they know its health benefits [17]. This pleasure is enhanced when combined with socialization [17].

With global competition between wine tourism regions intensifying, and the demand for wine tourism becoming more sophisticated, innovation and digital transformation in wine tourism are imperative [18]. The interest expressed in digital wine tourism arose from the fear and anxiety associated with the pandemic [19]. However, these feelings are not deterrents to the implementation of future wine tourism trips [19]. On the contrary, creating an asynchronous wine tourism experience, such as a virtual tour, could be a starting point for attracting new consumers in wine tourism [20]. In the post-pandemic era, companies and destinations are expected to redefine themselves to adapt to the digital age. The expansion of the digital presence of wineries also allows the most appropriate approach for wine tourists [21]. Wine producers can increase the value of a satisfying experience by bringing the winery closer to the customer, before, during, and after they visit the premises [22]. For example, the configuration of a winery in such a way that it gives the wine tourist the opportunity to take the "perfect Instagram photo" differentiates a winery from the rest and highlights the uniqueness of the wine destination [23].

Wine tourism in Greece is still at an early stage [14]. In a recent survey, only 22% of the surveyed Greek wineries had an online store with e-shopping [14]. At the same time, even though there is relevant national legislation for the certification of wineries that can have access to wine tourism services (Law 4276/2014) and corresponding promotions on a virtual platform, there is still no public list of these wineries accessible to visit in Greece, limiting their national support. Wine tourism in Greece, however, could be internationally supported, if requested, by the UN World Tourism Organization (UNWTO) which promotes strategies and protocols for the development of early-stage wine regions worldwide [24]. Respectively, the Global Wine Tourism Organization (GWTO) has provided scientific support towards European wine tourism by identifying, for example, the cost of living as the main determinant of tourism demand in Greece, followed by price competitiveness and income [25].

The GWTO considers wine tourism as a key contributor to the global economy with encouraging growth rates for the next decades [26]. Wine tourism addresses a niche market, which can be combined with all different types of tourism, thus promoting sustainable development for the local area [14]. Studies have shown that wine tourism can stimulate a rural local economy if small tourism businesses complement each other to form a strong local, sustainable destination image [27,28]. Therefore, the involvement of the oenological region has an important role in the sustainability of local wineries [29], thus highlighting

the two-way relationship between the local wine region and wine tourism. Many wine tourists, in fact, express great interest in the variety of attractions in the area that are not directly related to wine [30]. Visiting a winery from the perspective of Generation Z, for example, seems to be less about the wine itself and more about the opportunities to enjoy the scenery, have fun, socialize, and discover local products [31]. Wine tourists in Greece usually carry out additional activities in the wine region that are not related to wine after they visit the winery [32]. Particularly for Greek youth, research has shown that there is a limited appeal for the wine itself, with young people preferring activities such as enjoying the scenery and food as well as socializing as part of their visit to a winery [33]. This appeal also results from their limited financial ability to spend during their trip [33].

The recovery process from the pandemic suggests that domestic tourism is a definite driving force of the future. Recent research analyzing the elements of the tourist behavior of Greek citizens shows their strong preference for domestic tourism [34]. Significantly higher percentages of domestic visitors indicated a strong motivation to purchase wine during their winery visit, to chat with the winemaker or oenologist, and to participate in an organized winery tour or to experience the winery and atmosphere, in contrast to international visitors who preferred to participate in a group activity and take an excursion [35].

This is the first study that identifies and analyzes the overall consumer motives regarding wine tourism in Greece in the post-COVID-19 period. The purpose of this research is to evaluate those factors or parameters related to the motivations of domestic consumers to engage in wine tourism in a wine-producing country such as Greece and to identify the main objectives that the Greek wine tourism industry should focus on to ensure development in the post-pandemic era. To achieve this goal, and based on the existing literature on the related parameters of consumer preferences for wine tourism [36], the present study examines the following factors of engagement in Greek wine tourism during the post-COVID-19 period:

(I)     Wine consumption.
(II)    Existing experience in wine tourism.
(III)   Wine and winery parameters that encourage visits to a wine region.
(IV)    General characteristics of an oenological region that encourage a visit to a wine region.
(V)     Source of information consulted to encourage a visit to a wine region.

## 2. Materials and Methods

### 2.1. Data Collection and Sample Characterization

A survey was conducted based on a structured questionnaire that examined the various parameters that influence consumers' motivations for wine tourism in the post-COVID-19 era. The questionnaire was structured in six parts, formulated in such a way as to serve the purposes of the research, and was based on a previous study conducted in the past, in 2008, by Galloway et al. [36].

The first part consisted of five questions concerning the demographic data of the respondents, namely, gender, age, completed level of education, professional activity, and place of residence. The second part consisted of four questions concerning the consumption habits of the respondents regarding wine in the post-pandemic era which explores their connection with wine. The third part consisted of five questions concerning the respondents' experience with wine tourism. The fourth part consisted of 10 questions regarding the wine and winery parameters that would prompt the respondents to visit a wine region of Greece. The fifth part consisted of eight questions—parameters about the characteristics of the oenological region that encourage visits to a wine region. Finally, the sixth part consisted of six questions evaluating the importance of information sources that would encourage their involvement in wine tourism. Respondents were asked to rate each parameter from 1 (not at all important) to 5 (very important).

Table S1 presents the questionnaire, which was in electronic form (in Greek), through the Google Forms platform and was distributed via email. The email included an explanation of the purpose of the research, a brief interpretation of the concept of wine tourism,

as well as a link to the digital questionnaire. The answers were anonymous, and no personal data of the respondents was collected, in accordance with GDPR regulation and data protection. The structured questionnaire was distributed to the community of the University of Ioannina (students, professors, staff, etc.) through their academic emails (@uoi.gr), justifying the high percentage of young people and students in the sample. There were no specific criteria for the target group of the study.

The high percentage of women who answered the questionnaire (73.4%) is noteworthy, an observation that has been repeated in all previous research topics we have carried out that investigate consumer behavior [37–40], even though there were no significant gender-related taste differences on wine consumption detected [41].

The survey was conducted between April and May 2023, with the total number of participants being 595 and including a wide range of demographics.

### 2.2. Data Analysis

The analysis of the responses was carried out through basic statistical tools. The analytical data were organized using IBM SPSS Statistics for Windows (Version 25.0, IBM Corp. Armonk, NY, USA), following the methodology outlined by Skalkos et al. [39]. Cramer's V coefficient, which ranges from 0 to 1, was utilized in the chi-squared tests. Its interpretation is as follows: V values around 0.1 indicate a weak association, around 0.3 indicate a moderate association, and approximately 0.5 or higher indicate a strong association. For all conducted tests, a significance level of 5% ($p < 0.05$) was considered.

### 3. Results

Table 1 presents the demographic characteristics of the participants of the questionnaire.

**Table 1.** Demographic characterization of the sample.

| Variable | Groups | (%) |
|---|---|---|
| Gender | Male | 26.6 |
| | Female | 73.4 |
| Age | 18–25 | 67.9 |
| | 26–35 | 6.4 |
| | 36–45 | 5.4 |
| | 46–55 | 14.5 |
| | 56+ | 5.9 |
| Level of education (completed) | None/Primary School | 0.2 |
| | Secondary School | 0.7 |
| | High School | 53.8 |
| | University | 45.4 |
| Job situation | Unemployed | 1.3 |
| | Student | 67.4 |
| | Employed | 29.1 |
| | Retired | 2.2 |
| Permanent residence in Greece | North Greece (Macedonia and Thrace) | 25 |
| | West Greece (Epirus and Etoloakarnania) | 27.9 |
| | Central Greece (and Athens) | 34.6 |
| | South Greece (Peloponnese) | 4.9 |
| | Islands (Aegean, Ionian, and Crete) | 7.6 |

Regarding the place of permanent residence, 25% of the respondents came from Northern Greece, 27.9% from Western Greece, 34.6% from Central Greece (including the capital of Greece, Athens), while there was a significantly lower representation from Peloponnese and the islands of Greece with significant lower population (4.9% and 7.6%, respectively). Regarding the level of education, the majority held a high school or university diploma (53.8% and 45.4%, respectively), which is also linked to the fact that students

(67.4%) participated the most when regarding professional activity. More than 2/3 of the sample were young people (67.9%) and currently studying (67.4%), which can draw conclusions about the new generation and highlight the prospects for attracting them to wine tourism.

Table 2 presents the consumption habits of the respondents regarding wine. Most participants purchased 1–5 bottles of wine each month (60.2%), with a significant proportion reporting zero bottles of wine consumption (33.6%). Most respondents spent up to 10 EUR (52.8%) or 10–50 EUR (42.9%) on the purchase of wine monthly while the frequency of wine consumption had a smooth variation for once, twice, and three times weekly, except for the choice of daily consumption (2.9%). Noteworthy is the observation that most of the respondents stated that they consumed more wine after the pandemic (57%) compared with the period before.

**Table 2.** Wine consumption habits of the participants.

| Questions | Answers | (%) |
|---|---|---|
| How many bottles of wine do you buy per month TODAY? | Zero bottles/none | 33.6 |
| | 1–5 bottles | 60.2 |
| | 6–10 bottles | 5.2 |
| | 11–15 bottles | 0.5 |
| | >15 bottles | 0.5 |
| How much money do you spend on wine per month TODAY? | <10 EUR | 52.8 |
| | 10–50 EUR | 42.9 |
| | 50–100 EUR | 3.9 |
| | >100 EUR | 0.5 |
| How often do you consume wine? | Once per month | 28.1 |
| | Once every two weeks | 22 |
| | Once per week | 22.5 |
| | Twice per week | 24.5 |
| | Everyday | 2.9 |
| Do you consume LESS or MORE wine today compared to the pre-pandemic period? | Less | 43 |
| | More | 57 |

Table S2 depicts the significant ($p < 0.05$) associations between the respondents' consumption habits of wine based on their profile and sociodemographic variables. Specifically, many associations were found, with weak correlation on all the questions of *Part II* between age, level of education, and job situation. We also found that the question regarding the money spent per month on wine today was affected by gender as well, and the comparison regarding the consumption of wine before and after the pandemic was affected by the place of permanent residence.

Table 3 shows the results of the questions about the respondents' experience in wine tourism. Most respondents (59.8%) have visited a winery, out of which 32.6% have visited a winery once, 46.1% 2–3 times, and only 10.1% 4–5 times. Almost all the respondents who have participated in wine tourism (97.5%) have not stayed overnight at a winery (or partner accommodation). The most popular wine tourism activities they did during their visit were wine tasting (46.6%), followed by an open discussion about wine (35.2%), and buying wine at a discounted price (33%). At the same time, the most popular tourist activities carried out during their visit to the wine region were visiting the attractions of the region (46%), taking a tour of the region without a tour guide (31.5%), and activities in nature (30.3%).

**Table 3.** Wine tourism experience of the participants.

| Question | Answers | (%) |
|---|---|---|
| Have you ever visited a winery? | Yes | 59.8 |
| | No | 40.2 |
| If you answered YES to the first question, how many visits in the last 12 MONTHS? | 1 visit | 32.6 |
| | 2–3 visits | 46.1 |
| | 4–5 visits | 10.1 |
| | >5 visits | 11.2 |
| If you answered YES to the first question, did any of your visits include a stay at the winery? | Yes | 2.5 |
| | No | 97.5 |
| If you answered YES to the first question, select the oenological activities you did during your visit to a winery: | Wine tasting | 46.6 |
| | Buy wine at a discounted price | 33 |
| | Buy oenological literature | 1.3 |
| | Open discussion about wine | 35.2 |
| | Watch educational presentations | 23.2 |
| | Dinner | 8.2 |
| | Tour without a guide | 26.6 |
| | Art exhibition at the winery | 13.3 |
| | Try/buy local area products | 26.3 |
| | Picnic/BBQ | 5.9 |
| If you answered YES to the first question, select the tourist activities you did AT THE SAME TIME as your visit to a winery: | Visit area attractions | 46 |
| | Visit parks and recreation areas | 21 |
| | Dinner at local restaurants/ fine dining | 22.6 |
| | Activities in nature/outdoors | 30.3 |
| | Explore the history and culture of the area | 27.6 |
| | Visit local market | 27.4 |
| | Tour of the area WITHOUT a guide | 31.5 |
| | Guided tour of the area | 13.6 |

Table S3 presents the result of the chi-square test that revealed significant differences in the respondents' wine tourism experience based on their demographic characteristics. A medium association was found between the answers to the core question "Have you ever visited a winery" and age, with smaller ones regarding level of education (completed) and job activity. There were also many associations between age, job activity, and paid activities.

Table 4 represents the results of wine and winery parameters that would prompt participants to visit a Greek wine region as part of their wine tourism plans. Based on the positive choices (as quite and very important) (≥70%), the qualified staff (77.5%), followed by the wine regional reputation (76%), the existence of a winery tour and tasting (74.3%), festivals and special events (71.9%), as well as the cost of visit (70.4%) were the preferred parameters for a visit. The range of regional wineries (35.6%) was the lowest parameter of choice for visiting a region.

Table S4 depicts the result of the chi-square test that revealed small associations between the participants' answers on parameters regarding the wine and the winery and their demographic characteristics.

Table 5 presents the results of the parameters regarding the general characteristics of the oenological region that would encourage involvement in wine tourism. Based on the positive choices (as quite and very important) (≥70%), the existing natural environment (83.5%), the accessibility of the region (72.8%), and historical, cultural, and lifestyle regional opportunities (70.9%) were the most preferred parameters to encourage a visit. Surprisingly, the existence of gourmet cuisine was the least preferred parameter (37.2%). Parameters such as accommodation (58.3%), participation in social activities (55%), distance from home (53.4%), and the existence of a holiday program (50.1%) were of medium preference for visiting a region.

**Table 4.** Parameters regarding the wine and the winery that would encourage a participant's visit to a wine region in Greece.

| Parameters Regarding the Wine and the Winery | Not Important at All (1) | Slightly Important (2) | Important (3) | Quite Important (4) | Very Important (5) |
|---|---|---|---|---|---|
| Range of wineries in one region | 8.7 | 18.2 | 37.5 | 23.5 | 12.1 |
| Educational opportunities related to wine | 6.4 | 12.6 | 29.7 | 32.1 | 19.2 |
| Wine/wine tourism festivals and special events | 2.2 | 7.2 | 18.7 | 40 | 31.9 |
| Organized winery tour and local wine tasting | 1.8 | 7.1 | 16.8 | 38.7 | 35.6 |
| The existence of quality marks such as, e.g., PDO (Protected Designation of Origin) product, etc. | 2.9 | 9.7 | 19 | 30.1 | 38.3 |
| Area grape variety | 3.7 | 9.1 | 20.7 | 36 | 30.6 |
| Reputation of the wine region and the wine | 2.2 | 3.9 | 18 | 40 | 36 |
| Cost of visit and wine tasting | 2.5 | 5 | 22 | 32.6 | 37.8 |
| Staff knowledgeable about wine and the oenological region | 1.7 | 3.9 | 17 | 33.3 | 44.2 |
| Enrichment of knowledge about wine and grape varieties | 2.7 | 7.7 | 22.9 | 37 | 29.7 |

**Table 5.** Parameters regarding the general characteristics of the wine region to encourage visits to a wine region in Greece.

| Parameters Regarding the General Characteristics of the Wine Region | Not Important at All (1) | Slightly Important (2) | Important (3) | Quite Important (4) | Very Important (5) |
|---|---|---|---|---|---|
| Availability and variety of types of accommodation | 2 | 7.6 | 32.1 | 38.5 | 19.8 |
| Accessibility (airport, train station, and road accessibility) | 1.2 | 7.6 | 18.5 | 38.3 | 34.5 |
| Distance from the place of permanent residence | 3.2 | 14.1 | 29.2 | 30.4 | 23 |
| Personalized and organized holiday program (individual or group) | 5.4 | 13.4 | 31.1 | 35 | 15.1 |
| Opportunity to experience the culture, history, and lifestyle of the area | 1.5 | 6.6 | 21 | 36.3 | 34.6 |
| Availability of gourmet cuisine | 9.6 | 20.7 | 32.6 | 24.4 | 12.8 |
| Attractive natural scenery and good climatic conditions | 1.3 | 3.2 | 11.9 | 44 | 39.5 |
| Participation in social activities | 3.4 | 11.8 | 29.9 | 35 | 20 |

The results of the chi-square test that showed significant differences between consumers' opinions on the parameters regarding the general characteristics of the wine region to encourage wine tourism experience and their demographic characteristics are presented in Table S5.

Table 6 presents the results of the source of information requested by the participants in order to visit a region. Based on the quite and very important answers, information from social media (73%), and a recommendation by a third person (70.6%) were the preferred pieces of information requested for a visit, followed by the travel agent's recommendation (43.5%) and a recommendation by the local visitors' information center (42.7%). The least preferred data were from the media (35.6%) and brochures (23.6%).

**Table 6.** Importance of sources of information regarding wine tourism activities in Greece.

| Source of Information Regarding Wine Tourism Opportunities | Not Important at All (1) | Slightly Important (2) | Important (3) | Quite Important (4) | Very Important (5) |
|---|---|---|---|---|---|
| Brochures | 16 | 24.4 | 36.1 | 16 | 7.6 |
| Recommendation from a familiar person | 1.8 | 6.4 | 21.2 | 43.2 | 27.4 |
| Media (TV, and radio) | 8.1 | 18 | 38.5 | 25.7 | 9.7 |
| Local visitor information centers | 8.1 | 13.8 | 35.5 | 30.3 | 12.4 |
| Travel agencies—group visits | 8.9 | 15.1 | 32.4 | 29.9 | 13.6 |
| Internet/social media (Facebook, Instagram, TikTok,) etc. | 2.9 | 5 | 19.2 | 31.3 | 41.7 |

Table S6 presents the results of the chi-square test showing the significant differences between the participants' preference for the tool of information regarding wine tourism opportunities and their demographic characteristics.

## 4. Discussion

This study investigates, for the first time, the effect of the COVID-19 pandemic on consumer's motivations for wine tourism in a wine-producing region, such as Greece, to examine its development prospects in the post-pandemic period. The demographic characteristics of the participants depicted an increased percentage of young people, aged 18–25, like our other previous survey regarding quality wine consumption [40], which may highlight the desires and expectations of the next generation of wine tourists, as well as the path that wine tourism in Greece must take in order to attract them.

Regarding wine consumption habits, the participants, although they are frequent wine consumers, are not spending much on bottled wine, as they probably prefer bulk wine, which can be interpreted as a desire to buy locally produced wines, as proven by Di Vita et al. [42]. However, after the pandemic, there has been an increased consumption of wine compared to before, an observation that verifies a previous study by Pytell et al. [43].

Regarding the previous wine tourism experience of the participants, our findings indicate that the increased satisfaction of the consumer's wine tourism experience may increase their loyalty and encourage further engagement with wine tourism services (e.g., more than one winery visit), as recorded by other researchers [5,44]. The participants did not choose to use the accommodation options of the winery and instead were more driven to purchase wine after they visited the winery, as previously stated, and to invest in the local community, stimulating the economy of rural areas [45].

Regarding the motivations of the participants related to wine and the winery to keep them engaged with wine tourism, our results show that they were eager after COVID-19 to experience knowledge-sharing regarding the wine by qualified staff, rather than pleasure, as recorded by Bruwer et al. as well in 2019 (the pre-pandemic period) [46]. The young audience still seeks sustainable visit costs in the post-pandemic period, as suggested in 2018 by Stergiou et al. [33], with a primary interest in wine tourism festivals and special events. Such events can create new target groups for the wine tourism industry, as suggested by Yuan et al. first [47].

In terms of motivation regarding the wine region's amenities, we have shown that consumers seek a more sustainable wine tourism experience in the post-pandemic period, as shown already by Ouvrard et al. [48], with easy accessibility and nature-based activities. They are also willing to discover the history and culture of the region during their wine visit.

Regarding the sources of information for wine tourism opportunities, our results show that most participants consider it very important for the wineries to form a digital identity in the wine tourism industry, through the Internet and social media, highlighting the need for the inclusion of wine tourism in the digital world. Survey participants dodid not consider brochures or mass media to be important sources of information. These findings,

regarding the source of information sought for wine tourism selection and considering the high percentage of young (students of Generation Z) participants, are fully expected since the younger generations worldwide, more than other generations, have stopped reading brochures, maps, or watching TV as part of their daily habits.

## 5. Conclusions

This research paper investigates the motivations of consumers regarding their participation in wine tourism initiatives in wine-producing countries such as Greece, highlighting the parameters that will attract more domestic wine tourism in the post-pandemic period. The survey showed that although the participants consume more wine today compared to the period before the COVID-19 pandemic, the domestic demand for wine remains low (1–5 bottles, up to 10 EUR per month). Regarding the parameters for engagement in wine tourism, participants look for an organized winery experience, with trained staff and participation in events. Young Greeks are also looking for affordable wine tourism packages. The participants, moreover, showed that they are interested in exploring the local community when participating in wine tourism. Finally, they expect the wine industry to have all the proper information accessible online.

With the COVID-19 pandemic officially reaching its end, it is very important to analyze how it has affected human activity, what priorities it has now set, and how the wine tourism industry can ultimately adapt to today's needs. The wine tourism industry benefits from tourists' strong desire for sustainable tourism away from mass crowds. However, it is being tested with the need to enter the digital age and the competition between other forms of alternative tourism. The early stage of wine tourism in Greece is also an opportunity to adapt and evolve to new trends while ensuring its identity.

Wine tourism in wine-producing countries, based on the results of our work, needs three types of support to be able to develop further and successfully:

The first type is institutional support from the state, such as an institutional framework that ensures the quality of wine tourism services, highlighting wine wealth both in the context of representation and the context of digital promotion accessible to the public, and the creation of training programs for education and information about financing opportunities (e.g., "Leader" program of the EU).

The second type is support from the local community, including a collective effort made to be able to offer a complete wine tourism experience.

The third type is individual support from every winery in the country to create a comprehensive strategic plan to attract new consumers, to offer an enriched wine tourism experience, training, or hiring qualified staff, as well as innovation and the maintenance of a digital identity to be able to cope with the competition and evolve in the new trends that are created.

This study involved more women as well as an increased number of young students. Although it can be considered a limitation of the research, young Greeks will shape the future trends of domestic wine tourism and the results of this research can be used for this prospect. Another limitation is the participation of only Greek residents and not foreigners, who also shape the sustainability of wine tourism in Greece. A study based on an audience over 30 years old can highlight the desires needing to be satisfied by Greek wine tourism in the near future.

More studies, such as the impression that Greeks have of wine tourism today, their opinion on its future prospects, as well as a comparison with similar surveys in other EU states, will allow the formation of a more comprehensive view of how the pandemic has affected wine tourism, both in Greece and globally, and what challenges it has to face.

**Supplementary Materials:** The following supporting information can be downloaded at: https://www.mdpi.com/article/10.3390/su152316225/s1, Table S1: Questionnaire about wine tourism in Greece and its prospects of sustainable development in the post-pandemic period, Table S2: Associations between the consumption habits of wine and demographic variables, Table S3: Associations between participants' previous wine experience and demographic variables, Table S4: Associations

between respondents' answers on parameters regarding wine and wineries and their demographic variables, Table S5: Associations between the parameters regarding the general characteristics of the wine region to encourage participation in tourism activities in Greece and demographic variables, and Table S6: Associations between preference on source of information regarding wine tourism activities in Greece and demographic characteristics.

**Author Contributions:** Conceptualization and methodology, D.S. and A.S. writing—original draft preparation, A.S. and I.S.K. supervision and editing, D.S. All authors have read and agreed to the published version of the manuscript.

**Funding:** This research received no external funding.

**Institutional Review Board Statement:** Not applicable.

**Informed Consent Statement:** Not applicable.

**Data Availability Statement:** The data presented in this study are available on request from the corresponding author.

**Conflicts of Interest:** The authors declare no conflict of interest.

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
