# Peer review of "Consumers’ Motives on Wine Tourism in Greece in the Post-COVID-19 Era"

_sustainability, doi:10.3390/su152316225_

Round 1
Reviewer 1 Report
Comments and Suggestions for Authors
Dear Authors, Dear Editor,
The Authors of the manuscript "Consumers' motives on wine tourism in Greece in the new global era" present the results of their online survey of 595 respondents - Greek residents.
The research question should be considered important, especially nowadays, in a post-pandemic reality, as the tourism, food service and hospitality sectors are among the most affected industries. Parallel to mass tourism serving crowds of tourists each year in typical places (that must be seen and photographed), niche/specialty tourism is growing rapidly. It responds to the needs of tourists with concretized and special interests, for whom it is important to learn about culture and be active, where a small number of tourists reach authentic places. An example of such tourism is food tourism, culinary tourism or gastronomic tourism (gastrotourism). For those experiencing this tourism, it is important to learn about unique local and regional intangible cultural heritage. A type of niche tourism is wine tourism and this topic is explored by the Authors using Greece as an example.
The purpose of the study was to identify factors that are important to consumer-tourists and potentially influence the development of wine tourism in major wine-producing countries.
The reviewed work has application value, as it can inspire innovative expansion of the offer for winery operators. It can make an important contribution to the literature on consumer perspectives on wine tourism, provided that three aspects are strengthened.
First, the literature review or introduction should be supplemented with a picture of the global/European wine tourism sector, using, among other things, data and information from international organizations GWTO and UNWTO, as well as national ones, such as in Greece, France, etc.
Second, the description of the survey methodology needs to be supplemented and at the same time shortened. More details on the selection of respondents should be provided. What database of email addresses was used? - this is especially important in the context of the sentence in L. 194-197, which should be included in the methodology. Were there established criteria for entry into the study (e.g., age, wine drinking experience(?). Was there any assumption made about a particular characteristic(s) of the respondents? What determined the sample size? In subsection 2.2, leave only the first sentence and the text in L. 175-181. There is no point in repeating the substantive parts of the questionnaire, since they have already been presented on the previous page, and the socio-demographic characteristics of the respondents are obvious in the presentation of consumer research. The next paragraph describing the structure of the questionnaire and the types of questions (uncomplicated) is also superfluous, since the questionnaire in its original version was included as a supplement. It is worth noting that the question "Do you consume LESS or MORE wine today compared to the pre-pandemic period?" is a forcing question, since it does not include the "same amount" option (but this can no longer be fixed).
Third, the Discussion section needs to be rewritten, rethought and deepened. The Authors use the word "more" (L. 402, 404, 412) unjustifiably, since they do not refer in any way to the results of the pre-pandemic study. In the study presented here, we have the current picture of consumer motivation, and this one should be discussed. For the same reason, it is unjustified to state in L. 394-396 "our findings indicate that the increased satisfaction of the consumer's first wine tourism experience still tends to increase their loyalty and encourage further engagement." - the questionnaire does not ask about the first eno-tourism experience. The assumption in the last sentence of this section is also wrong, as Generation Z (nearly 68% of respondents) and young adults of the millennial generation (nearly 12%) stopped reading brochures, maps and watching TV long before the pandemic. I leave it to the Authors to address the very young age of the respondents (67.9% declared an age group of 18-25) in the context of alcohol consumption.
I have a few more minor suggestions for Authors to consider.
1. In the title of the article instead of "in the new era" it would be better to write "in the post-covid era" or, for example, "in the post-covid reality".
2. As the last key word, it is enough to write "Greece," since "wine tourism" is already listed as the second.
3. Paragraph L. 188-194 can be improved by showing the general characteristics of the sample (e.g., 2/3 of the sample are young people, well educated (99%), students ... etc.), since these precise data are included in Table 1.
4. From the description of Table S2, all the detailed data from the results of the statistical analysis should be removed, because all these data are in this table. Here you can only leave the questions and the characteristic(s) that determined the responses. However, it is worth adding that these were only weak correlations (given the values of the Cramer coefficient).
5. Identically, the description of Table S3 should be changed (i.e., remove all data from the analysis). Due to the large number of questions, the effects of this analysis can be presented in a table, where in column 1 will be the questions, and in the following columns all the characteristics as explanatory variables. Where there was a relationship, the value of the Cramer coefficient can be given.
6. And in the same way, remove all details from the description of Tables S4, S5, S6, evaluating the strength of the relationship according to the value of the Cramer coefficient.
7. And 3 more spelling errors that I managed to catch: the last question in Tab. 2 - should be "consume", the name of column 3 in Tab. 4 - should be "important", L. 382 - remove the Greek letter alpha.
Kind regards
Reviewer 2 Report
Comments and Suggestions for Authors
Overall, an interesting topic has been presented. My main comment has to do with making the gap in literature more evident to support how your work contributes to the existing literature.
More comments on your paper that I believe should be considered to improve the quality of your work:
- The introduction presents the topic to the reader and concepts are introduced. However at line 35 I would expect a more direct link between tourism/sustainability (lines 31-35) and the wine consumption or maybe rephrasing the existing one to link it more direct.
- At the end of the introduction the structure of the paper should be presented to the reader.
-The literature sets the scene for the empirical analysis and the situation in Greece is well presented. However, I would expect the gap in the existing literature to be more evident and clarified.
- In the materials and methods section you nicely comment on the gender imbalance however I would expect you to justify the sample being skewed towards women. Maybe there is evidence in literature showing that wine is primarily consumed from women? or the topic of the survey attracted more women.
- Moreover, you need to state whether the questionnaire was a semi-structured or structured one, the sampling method employed to collect the sample, whether a pilot study has been conducted and what changes have been made after as well as how did you ensure reliability and validity.
-Results are well presented and appropriate statistical techniques have been applied. These could be expanded, in case you wish that though, for example multiple linear regression could be used to examine the impact of several factors or of the statistically significant factors found in the current analysis on wine consumption.
-Discussion part discusses findings with existing literature comparing and contrasting. The conclusion presents the main highlights of the work and policy implications are presented along with recommendations. Moreover, the limitations of this work have been acknowledged along with further research proposed.
Round 2
Reviewer 1 Report
Comments and Suggestions for Authors
Dear Authors,
I am satisfied with the corrections you have made to the text according to my suggestions. All my suggestions have been taken into account and I am satisfied that I could contribute to enhancing the scientific value of your article.
I guess I can already congratulate you on your new publication?
Kind regards